# Diagnostic and Clinical Outcomes of Three Regenerative Strategies for Alveolar Bone Defects: A Comparative Study Using CBCT and ISQ

**DOI:** 10.3390/diagnostics15162078

**Published:** 2025-08-19

**Authors:** Sorin Gheorghe Mihali, Șerban Talpoș, Mălina Popa, Dan Loloș, Serafina Bonomo, Tareq Hajaj

**Affiliations:** 1Department of Prosthodontics, Faculty of Dentistry, “Vasile Goldiș” Western University of Arad, 94 Revoluției Blvd., 310025 Arad, Romania; mihali.sorin@uvvg.ro; 2Discipline of Oral and Maxillo-Facial Surgery, Faculty of Dental Medicine, “Victor Babeș” University of Medicine and Pharmacy Timișoara, Revoluției Boulevard 9, 300041 Timișoara, Romania; 3Department of Pediatric Dentistry, Faculty of Dental Medicine, “Victor Babeș” University of Medicine and Pharmacy, Eftimie Murgu Square 2, 300041 Timișoara, Romania; 4Faculty of Dental Medicine, “Victor Babeș” University of Medicine and Pharmacy Timișoara, Eftimie Murgu Square No. 2, 300041 Timișoara, Romania; lolosdan@umft.ro; 5Faculty of Dental Medicine, “Vasile Goldiș” Western University of Arad, 94 Revoluției Blvd., 310025 Arad, Romania; serafina98@icloud.com; 6Discipline of Prostheses Technology and Dental Materials, Faculty of Dental Medicine, “Victor Babeș” University of Medicine and Pharmacy, Eftimie Murgu Square 2, 300041 Timișoara, Romania; tareq.hajaj@umft.ro

**Keywords:** platelet-rich fibrin, guided bone regeneration, implant stability, bone graft, collagen membrane, soft tissue healing

## Abstract

**Background**: This prospective clinical study aimed to evaluate the effectiveness of platelet-rich fibrin (PRF) in guided bone regeneration (GBR) prior to dental implant placement. **Material and methods**: Sixty-five patients with alveolar bone defects were randomly assigned to three groups. All groups received a composite graft consisting of 70% allograft and 30% xenograft. Group A received the graft combined with PRF. Group B received the graft with PRF and a resorbable collagen membrane. Group C (control) received the same graft and membrane without PRF. Cone-beam computed tomography (CBCT) was used to assess bone regeneration at baseline and 6 months postoperatively. Implant stability was evaluated using ISQ values at the time of implant placement (6 months after grafting) and again at 3 to 4 months during the second-stage uncovering procedure. Soft tissue healing, postoperative complications, and pain scores were also recorded. **Results**: Group B showed the best outcomes, with the highest mean vertical bone gain (3.0 ± 0.4 mm), greatest implant stability (ISQ: 74.2 ± 1.8), and no complications. Group A achieved moderate bone gain (2.3 ± 0.4 mm) and good ISQ values (71.5 ± 2.3), with favorable soft tissue healing. In contrast, Group C had the lowest bone gain (2.1 ± 0.5 mm), reduced ISQ values (68.9 ± 2.9), and the highest incidence of complications, including dehiscence and minor infections. **Conclusions**: These results suggest that PRF enhances both hard and soft tissue regeneration, particularly when used with grafts and membranes. PRF may reduce healing time and postoperative discomfort, improving the overall success of regenerative implant procedures.

## 1. Introduction

Guided bone regeneration (GBR) is a well-established surgical technique used to restore alveolar bone volume lost due to atrophy, trauma, systemic conditions, or tumor resection [1,2]. It is widely employed to create a suitable bone foundation for implant placement by guiding cellular proliferation within osseous defects [3,4]. The process relies on barrier membranes and bone graft materials that promote the migration and activity of osteogenic cells while excluding undesired soft tissue cells [5,6].

Bone grafts are generally classified into four main categories based on their origin: autogenous (harvested from the same individual), allogeneic (from a donor of the same species), xenogeneic (from a different species), and alloplastic (synthetic substitutes) [7,8]. Table 1 summarizes the most frequently used materials, along with their advantages and disadvantages relevant to clinical decision making.

The success of GBR also depends on the properties of the barrier membrane. Non-resorbable membranes, such as PTFE, are effective but require a second surgical procedure for removal [9,10]. Resorbable membranes, usually made from collagen, improve healing dynamics and patient comfort [11,12,13]. To further enhance regeneration, autologous platelet concentrates have gained increasing interest.

**Table 1 diagnostics-15-02078-t001:** Classification of bone grafting materials by origin.

Type of Bone Graft	Origin	Examples	Advantages	Disadvantages
**Autologous**	From the same individual	Iliac crest bone, mandibular bone	Osteogenic, osteoinductive, no immune rejection [9,11]	Limited quantity, morbidity, requires further intervention [7,9]
**Allogenic**	From a different individual of the same species	Freeze-dried bone allograft, demineralized bone matrix (DBM)	Available in larger quantities, no donor site morbidity [9,11]	Weakly osteoinductive, infective risks, costs, immune response [10,11]
**Xenogenic**	From a different species	Bovine-derived bone grafts (e.g., Bio-Oss)	Osteoconductive, widely available, cheap [9]	Possible immune reaction, slower integration [9,10]
**Alloplastic**	Synthetic substitutes	Hydroxyapatite, tricalcium phosphate, bioactive glass	Unlimited supply, no disease transmission risk [9,11]	Variable price, ability to act as a foreign body [7]

Platelet-rich fibrin (PRF) is a second-generation platelet concentrate obtained through a simple centrifugation process without anticoagulants. It forms a fibrin matrix enriched with platelets, leukocytes, cytokines, and growth factors that are released gradually, supporting tissue repair over an extended period [14,15,16,17]. Compared to platelet-rich plasma (PRP), PRF provides a longer release of growth factors, promotes angiogenesis, reduces postoperative pain, and lowers the risk of infection [18,19,20].

PRF enhances bone and soft tissue regeneration through both biological and structural mechanisms. It contains a dense fibrin network that traps platelets, leukocytes, and high concentrations of growth factors such as platelet-derived growth factor (PDGF AA and PDGF BB), transforming growth factor beta 1 (TGF β1), vascular endothelial growth factor (VEGF), insulin-like growth factor, and fibroblast growth factor 2. These mediators are released gradually over a period of 10 to 15 days, which prolongs their stimulatory effect compared with platelet-rich plasma [17,21]. PDGF and TGF β1 are involved in the recruitment of mesenchymal stem cells and osteoblasts, promoting their migration, proliferation, and differentiation. TGF β1 also regulates the inflammatory response and supports the formation of extracellular matrix, which stabilizes the healing site [22]. VEGF contributes to the formation of new blood vessels, improving oxygen and nutrient delivery to the regenerating tissues [23].

The fibrin matrix functions as a natural scaffold that facilitates osteoblast adhesion and cell migration, providing a supportive structure for tissue repair [24]. PRF also stimulates several intracellular signaling pathways such as Smad, ERK1/2, PI3K/Akt, and Wnt/β catenin, which are associated with osteogenic differentiation and bone formation [25]. In addition to these regenerative properties, PRF contains leukocytes and antimicrobial peptides that help control infection and modulate the local immune response, improving postoperative outcomes [24]. Recent studies have shown that PRF improves horizontal ridge width and reduces graft resorption in guided bone regeneration procedures [26]. Titanium-prepared PRF has demonstrated better ridge preservation, higher bone density, improved soft tissue healing, and less postoperative pain compared with conventional PRF or spontaneous healing [27].

By supporting cell recruitment and differentiation, providing a stable scaffold, and maintaining a sustained release of bioactive factors, PRF enhances tissue regeneration and represents a valuable autologous adjunct in guided bone regeneration protocols. However, it lacks the mechanical stability to maintain space in large defects, which requires its combination with particulate grafts or membranes for predictable outcomes [28].

PRF stimulates both soft and hard tissue healing by recruiting mesenchymal cells, enhancing osteoblast proliferation, and releasing angiogenic factors such as VEGF, which improves vascularization at the surgical site [29,30,31]. Its fibrin scaffold also acts as a reservoir for signaling molecules and facilitates the adhesion and migration of osteoblasts and endothelial cells, ultimately promoting new bone formation [32]. Despite its biological advantages, PRF must be centrifuged immediately after blood collection because of rapid coagulation, which allows only a narrow processing window [33].

In guided bone regeneration, PRF has shown promising results as an adjunctive biomaterial, particularly when combined with graft materials and resorbable membranes. It improves healing time, enhances peri implant soft tissue quality, and contributes to better esthetic outcomes [34,35]. Nevertheless, some studies report that PRF alone may be sufficient only for minor defects, while combined approaches yield better results in complex clinical scenarios [36].

This study aims to evaluate the clinical performance of PRF in GBR procedures, focusing on its effects when used alone or in combination with bone grafts and membranes. The study also assesses implant stability, osseointegration, tissue healing, and peri-implant soft tissue integration to determine whether PRF can improve the prognosis of implant treatments.

## 2. Materials and Methods

This prospective, randomized controlled clinical study was carried out between April 2023 and September 2024 and involved a total of 65 patients requiring bone augmentation procedures prior to dental implant placement. All procedures were conducted at the private dental clinic Dental Concept by Dr. Mihali (Timișoara, Romania), under standardized surgical conditions. The study was conducted in accordance with the principles outlined in the Declaration of Helsinki, with ethical approval obtained from the Ethics Committee of the University of Medicine and Pharmacy “Victor Babeș” of Timișoara, Romania (Approval Nr. 85/04.04.2023 rev 2025). All participants were informed in advance about the nature of the study and signed written consent forms. Patient privacy and safety were fully protected throughout the research process.

Patient selection was based on clearly defined inclusion and exclusion criteria. Eligible patients were aged between 30 and 65 years, systemically healthy, non-smokers, and presented with horizontal or vertical alveolar bone defects necessitating guided bone regeneration (GBR). Patients were excluded if they were active smokers, suffered from uncontrolled diabetes or systemic inflammatory diseases, had a history of bisphosphonate therapy or radiation treatment in the head and neck region, or showed poor oral hygiene at the time of evaluation.

Patients were randomly allocated into three treatment groups. All groups received a bone graft composed of 30% xenograft (Bio-Oss®, Geistlich Pharma AG, Wolhusen, Switzerland) and 70% allograft (Puros Cancellous Particulate Allograft, ZimVie, Westminster, CO, USA).

For better visualization of the study protocol, patient selection, randomization, group allocation, surgical procedures, and postoperative follow-up are summarized schematically in the flowchart presented in Figure 1.

Group A received the bone graft in combination with autologous platelet-rich fibrin (PRF).

Group B received the bone graft combined with PRF and covered with a resorbable collagen membrane (Biomed Extend, Zimmer Biomet, Warsaw, IN, USA).

Group C (control) received the same bone graft and membrane as Group B, but without the use of PRF.

Following bone augmentation, a provisional restoration was placed in all cases to preserve the augmented space, support soft tissue healing, and shield the surgical area from mechanical trauma.

In our study, PRF was prepared by collecting 10 mL of peripheral venous blood from each patient into sterile, additive-free vacuum tubes. The samples were immediately centrifuged at 1500 revolutions per minute (RPM) for 14 min using the Plasmalifting™ XC-2415 Series centrifuge (Plasmalifting Technologies LLC., Moscow, Russia). This protocol, applied uniformly across all cases requiring PRF, resulted in the formation of a fibrin clot located between the red blood cell layer and the platelet-poor plasma. The clot was carefully extracted and compressed in a sterile PRF box to produce a membrane suitable for surgical application (Figure 2).

Additional reference protocols for alternative PRF variants are summarized in Table 2 for comparative purposes.

All surgical interventions were performed under local anesthesia, following preoperative medication and standard sterile field preparation. A crestal incision was made, followed by full-thickness mucoperiosteal flap elevation, with subperiosteal detachment extending at least 5 mm beyond the defect margins. After complete exposure, the bone defect was thoroughly debrided of granulation tissue and fibrous remnants. In cases with dense cortical bone, decortication was performed by creating multiple perforations with a small round bur, facilitating medullary bleeding and improving cell recruitment and vascularization.

In Group A (PRF + Composite Graft − 70% Allograft + 30% Xenograft), bone defects were first filled with the composite graft, followed by the application of autologous PRF membranes directly within and over the defect. The membranes were sutured to the periosteum to prevent displacement and to support localized angiogenesis. No additional barrier membrane was used in this group, as shown in Figure 2.

In Group B (PRF + Composite Graft + Collagen Membrane), the bone defects were filled with a composite graft consisting of 70% allograft and 30% xenograft, covered with a resorbable collagen membrane (Biomed Extend, Zimmer Biomet, Warsaw, IN, USA). Before flap closure, multiple layers of autologous PRF membranes were placed over the membrane to enhance vascularization, support graft integration, and reduce postoperative inflammation, as illustrated in Figure 3 and Figure 4.

In Group C (Composite Graft + Collagen Membrane − Control group), the bone defect was filled with a composite graft consisting of 70% allograft and 30% xenograft. A resorbable collagen membrane (Biomed Extend, Zimmer Biomet, Warsaw, IN, USA) was applied over the graft and stabilized to the periosteum using resorbable sutures. No PRF membranes were used in this group as seen in Figure 5 and Figure 6. Flap closure was performed with a combination of deep horizontal mattress sutures to eliminate tension and single interrupted sutures at the epithelial level for precise adaptation.

Postoperative care was standardized for all patients. Systemic antibiotics were prescribed for six days: amoxicillin–clavulanate (Augmentin^®^, 1 g, GlaxoSmithKline, Brentford, UK) or, in penicillin-allergic patients, clindamycin (Dalacin C^®^, 300 mg, Pfizer, New York, NY, USA). Oral analgesics included ketoprofen (Oki^®^, 100 mg sachets, Dompé, Italy), administered as needed. Patients were instructed to use 0.12% chlorhexidine mouthwash twice daily for two weeks and to avoid any mechanical trauma or pressure on the grafted site. Sutures were removed between 10 and 14 days postoperatively, based on individual healing progression. Clinical and radiographic follow-up was scheduled at 6 and 12 months post-surgery.

Cone-beam computed tomography (CBCT) scans were performed at baseline and at 6 months to evaluate the horizontal and vertical bone gain, cortical thickness, and trabecular density. Implant stability was assessed using resonance frequency analysis (RFA) with the Osstell™ ISQ device (Osstell AB, Gothenburg, Sweden). Implant Stability Quotient (ISQ) values were recorded at the time of implant placement (approximately 6 months after grafting) and again at 3 to 4 months during the second-stage uncovering procedure.

Cone-beam computed tomography (CBCT) evaluations were performed for each case using the Planmeca Romexis^®^ 3D Imaging Software (Planmeca, Helsinki, Finland). The scans allowed multiplanar reconstructions, including axial, panoramic, and cross-sectional views, as well as three-dimensional (3D) volumetric renderings for comprehensive evaluation of the augmented sites. For each patient, CBCT cross-sectional analysis was guided by standardized anatomic landmarks. While slice positions varied slightly depending on individual anatomy, matched pre- and postoperative sections were selected consistently to allow accurate comparison of bone dimensions and trabecular density.

To ensure reproducibility, each patient underwent CBCT scanning in a standardized seated position. The region of interest (ROI) was identified based on adjacent anatomical structures, and horizontal ridge width was measured at the same location before and after grafting. Bone density values were recorded in Hounsfield Units (HU) using calibrated Romexis^®^ software tools, and the presence or absence of cortical continuity, dehiscence, or fenestration was also assessed. This approach enabled both dimensional and histo-radiologic evaluation of graft integration.

A representative case is shown in Figure 6 to illustrate the CBCT-based evaluation protocol. Cross-sectional slices between 45.2 and 46.0 mm were analyzed using the same acquisition settings and patient positioning as in all other cases. Radiographic measurements were performed by a calibrated examiner using standardized tools. This example demonstrates the methodology used to assess ridge width, trabecular density, and cortical continuity.

All scans were acquired with identical device settings and patient positioning, and all radiographic measurements were performed by a calibrated examiner as seen in Figure 7.

All CBCT scans were acquired using a standardized protocol with a field of view (FOV) of 5 × 5 cm for localized defects and 11 × 5 cm for extended ridge defects. The voxel size ranged between 150 μm and 200 μm depending on the selected FOV. Ultra-low dose (ULD) mode was activated for all scans to minimize radiation exposure. The acquisition time was approximately 4–5 s, with a tube voltage of 86–89 kV and a tube current of 6.3–8.0 mA.

Patients were randomly allocated to the three study groups using a computer-generated randomization list prepared by an independent investigator who was not involved in the surgical procedures or outcome assessment. Group assignments were concealed in sequentially numbered opaque envelopes that were opened only at the time of surgery to ensure allocation concealment. All CBCT-based radiographic measurements and ISQ recordings were performed by a single calibrated examiner who was blinded to the treatment group allocation. To ensure reproducibility and minimize intra-observer variability, each radiographic measurement was repeated twice at a two-week interval. Intra-examiner agreement was calculated using the intraclass correlation coefficient (ICC), which demonstrated excellent reliability (ICC = 0.94).

The remaining CBCT-based evaluations and their respective clinical interpretations are presented in the Results section.

## 3. Results

A total of 65 patients were enrolled in the study, including 34 males (52.3%) and 31 females (47.7%), with a mean age of 46.3 ± 7.8 years (age range 30–64 years, median 47 years). The procedures were performed in both the maxilla and the mandible, covering all dental regions. In the maxilla, 20 patients (30.8%) required interventions in the anterior region (incisors and canines), 10 patients (15.4%) in the premolar area, and 5 patients (7.7%) in the molar region. In the mandible, 8 patients (12.3%) presented indications in the anterior region, 12 patients (18.4%) in the premolar area, and 10 patients (15.4%) in the molar region. This distribution reflects the predominant esthetic demands in the anterior maxilla, as well as the functional and occlusal rehabilitation needs more frequently encountered in the mandible.

General demographic data are summarized in Table 3. Participants were evenly distributed among the three study groups: Group A (PRF only, n = 20), Group B (PRF combined with 70% allograft, 30% xenograft, and a resorbable membrane, n = 25), and Group C (control group receiving graft plus resorbable membrane without PRF, n = 20). Detailed demographic characteristics and group distribution are presented in Table 4.

Bone regeneration was assessed using cone-beam computed tomography (CBCT), with measurements obtained preoperatively and at 6 months postoperatively. The baseline mean bone height was approximately 2.9 mm across all groups. At the 6-month follow-up, CBCT analysis showed a mean bone height of 5.2 ± 0.3 mm in Group A, 5.8 ± 0.4 mm in Group B, and 5.1 ± 0.4 mm in Group C. This translates into a mean vertical bone gain of 2.3 ± 0.4 mm for Group A, 3.0 ± 0.4 mm for Group B, and 2.1 ± 0.5 mm for Group C. As illustrated in Figure 8, the highest regenerative outcome was recorded in Group B.

Statistical analysis using one-way ANOVA followed by Tukey’s post hoc test confirmed that the difference between Group B and the other two groups was statistically significant (*p* < 0.01). Implant stability, assessed via ISQ (Implant Stability Quotient), showed improvement in all three groups from the moment of placement to the 3-month re-evaluation. At placement, Group A recorded a mean ISQ of 65.4 ± 3.8, while Group B and Group C reported 70.1 ± 3.1 and 64.2 ± 4.0, respectively. After three months, ISQ values increased to 71.5 ± 2.3 for Group A, 74.2 ± 1.8 for Group B, and 68.9 ± 2.9 for Group C. Group B again demonstrated the most favorable evolution, with statistically significant differences compared to the other groups (*p* < 0.01). These dynamics are presented in Table 5.

Soft tissue healing was consistently described as clinically stable and uneventful in the PRF-treated groups. Both Group A and Group B exhibited favorable healing patterns, with no recorded instances of wound dehiscence or postoperative infection. Clinical notes frequently described healing as “excellent” and “stable”. In contrast, Group C displayed a 25% incidence of wound dehiscence (5 out of 20 patients), along with minor infections in 15% of cases (3 patients), all of which resolved with conservative antibiotic therapy. Notably, two patients in Group C experienced graft resorption exceeding 20% of the initially placed volume. To explore potential associations between key clinical outcomes, additional correlation analyses were conducted. A statistically significant moderate positive correlation was identified between vertical bone gain at 6 months and ISQ values recorded at the 3-month follow-up (r = 0.49; *p* < 0.001), suggesting that greater regenerated bone volume is associated with improved implant stability as seen in Figure 9.

Furthermore, a statistically significant positive correlation was identified between total healing time and the presence of postoperative complications (r = 0.40; *p* = 0.001), indicating that complications such as infection or dehiscence tend to prolong the healing process, as seen in Figure 10.

To evaluate the significance of differences in infection rates among the three groups, a Chi-square test was performed. The analysis yielded a Chi-square statistic of 7.08 with 2 degrees of freedom and a *p*-value of 0.029. This result indicates a statistically significant association between treatment groups and the incidence of postoperative infection. The highest frequency of infections was observed in Group C, which did not receive PRF treatment. These data are graphically summarized in Table 6.

Postoperative recovery was further assessed through VAS (Visual Analog Scale) pain scores and overall healing duration. Group B patients reported the lowest pain intensity on the first postoperative day, with a mean VAS of 2.1 ± 0.6 and a healing period of 5.8 ± 0.5 weeks. Group A followed with a VAS of 3.0 ± 0.7 and a healing time of 6.4 ± 0.6 weeks, while Group C reported the highest pain intensity (VAS 3.8 ± 0.5) and the slowest recovery (7.2 ± 0.8 weeks). These comparisons are illustrated in Figure 11.

The results indicate consistent trends across all assessed parameters, with Group B demonstrating the most favorable outcomes in terms of bone gain, implant stability, healing time, and postoperative complications.

Cone-beam computed tomography (CBCT) analysis was used to quantitatively assess dimensional changes and bone quality at the augmentation sites. Postoperative scans at 6 months were compared to baseline images using matched cross-sectional slices, ensuring high precision in measuring ridge width and trabecular density.

In a representative case, cross-sectional evaluation was focused on the 42.4 mm slice (Figure 12). Preoperatively, the alveolar ridge presented a critical horizontal deficiency, with a width of only 1.20 mm and thin, discontinuous cortical plates. The average trabecular bone density was 777.10 HU, with values ranging from 22 to 1295 HU, indicating predominantly low-density cancellous bone (Figure 12a).

At 6 months post-augmentation, the same slice showed a ridge width of 11.63 mm, demonstrating a net horizontal bone gain of 10.43 mm. Radiographically, the area displayed continuous corticalization and homogeneous trabecular density, confirming effective mineralization of the grafted site (Figure 12b).

Three-dimensional renderings were also employed to evaluate volumetric changes and spatial morphology of the mandibular bone (Figure 13). Before augmentation, the 3D reconstruction revealed severe horizontal and vertical atrophy in the posterior mandible. The cortical outline was thin, irregular, and interrupted, with insufficient bone volume for implant placement (Figure 13a).

Postoperatively, the augmented ridge exhibited continuous, well-defined cortical borders and substantial volumetric gain in both horizontal and vertical dimensions. The reconstructed segment appeared anatomically stable and suitable for implant placement (Figure 13b).

In another case, CBCT assessment of the posterior maxilla confirmed successful vertical and horizontal regeneration (Figure 14). At 6 months postoperatively, the bone height from the alveolar crest to the sinus floor measured 13.79 mm, while the bucco-palatal ridge width reached 6.80 mm. The mean trabecular density was 685.74 HU, with a density range of 148 to 1165 HU, indicating well-mineralized trabecular bone and developing cortical structures. The radiologic findings demonstrated excellent graft integration and volume preservation, sufficient bone height and width to allow safe and predictable implant placement, and favorable bone quality in a region traditionally considered challenging due to low baseline density.

## 4. Discussion

The results of this prospective clinical trial highlight the beneficial role of platelet-rich fibrin (PRF) in guided bone regeneration (GBR), particularly when combined with xenogeneic and autogenous bone grafts and a resorbable membrane. This combination led to superior outcomes in terms of bone volume gain, implant stability, and soft tissue healing. These findings are consistent with previous studies emphasizing the regenerative capacity of PRF as an autologous source of growth factors and a three-dimensional fibrin matrix that facilitates cell migration, angiogenesis, and tissue repair. In one study, using PRF alongside graft materials and membranes enhanced clinical outcomes in bone augmentation procedures, supporting the synergistic effects observed in our Group B [17].

Regarding bone regeneration and implant stability, Group B exhibited the highest gain in bone height and the most favorable ISQ values. This supports the notion that PRF, when used in conjunction with structured grafts, not only provides biological stimulation but also contributes to mechanical integrity. These observations are consistent with findings reporting that PRF supports osteoblast proliferation and bone integration due to its gradual release of growth factors over a 7–10-day period [32,35].

Although Group A (PRF alone) showed improved soft tissue healing and moderate bone gain, the lack of a mechanical scaffold limited its effectiveness in larger defects. This confirms previous reports highlighting the importance of space-maintaining materials in achieving predictable bone regeneration [10]. One recent meta-analysis also found that PRF application in ridge augmentation significantly enhances horizontal bone gain, promotes soft tissue regeneration, and reduces postoperative discomfort, although these conclusions were drawn from a relatively limited number of studies, emphasizing the need for further clinical validation [26].

A systematic review comparing PRF and concentrated growth factors (CGFs) combined with different scaffold materials concluded that PRF positively influences sinus lift procedures, ridge preservation, and defect regeneration while reducing bone resorption and promoting soft tissue healing. However, not all included studies showed statistically significant benefits, highlighting the variability of results and the necessity for further investigation [37]. Another meta-analysis investigating PRF’s influence on implant stability revealed that PRF application significantly improved primary and secondary stability, supporting faster bone healing and enhanced osseointegration [38]. These conclusions were further supported by a review using resonance frequency analysis (RFA) via Osstell measurements to confirm the improved implant stability associated with PRF [39].

Taken together, these reports reinforce the evidence that PRF, particularly when used in combination with grafting materials and membranes, can enhance both hard tissue regeneration and implant stability. Clinically, this translates into a more predictable and biologically favorable healing process. With regard to soft tissue healing and postoperative morbidity, Groups A and B demonstrated faster healing, less discomfort, and an absence of postoperative complications such as dehiscence or infection. In contrast, the control group, Group C, exhibited a 25% rate of dehiscence and a 15% incidence of minor infections. This contrast further supports the relevance of PRF in minimizing soft tissue complications. Several prior studies have pointed in this direction. For example, a review from 2019 demonstrated PRF’s ability to reduce postoperative pain, while more recent investigations observed that PRF application in extraction sockets not only reduced discomfort but also preserved more ridge volume compared to spontaneous healing [28,33]. Furthermore, the application of PRF after mandibular third molar removal was found to significantly reduce the incidence of dry socket, postoperative swelling, and trismus, as well as accelerating soft tissue regeneration and bone density improvements [40,41].

These findings are also consistent with previous reports indicating that PRF use reduces healing complications and enhances graft integration [42]. Additionally, CBCT-based evaluations, as adopted in our protocol, have been validated as reliable tools for assessing bone augmentation outcomes [42]. A wide range of systematic reviews and meta-analyses continue to underscore the beneficial effects of PRF in treating periodontal defects and furcation lesions, especially when used alongside bone substitutes and membranes, further confirming the reproducibility of our results in broader clinical applications [43,44].

Despite these encouraging findings, our study does present several limitations. First, the follow-up period was limited to 12 months, preventing long-term assessments of bone stability and implant success. Second, being conducted at a single private clinic, the study lacks geographic and operator diversity. Third, no histological evaluation was performed to assess bone maturation at a microscopic level. Lastly, potential operator-dependent variables were not statistically adjusted, which could introduce minor biases in surgical outcomes.

Future research should address these limitations by incorporating long-term, multicenter trials and histomorphometric analyses. Furthermore, exploring variations of PRF, such as advanced-PRF (A-PRF) or injectable-PRF (i-PRF), may help determine optimal formulations and protocols suited to different clinical scenarios. Given its autologous nature, low cost, and minimal risk profile, PRF remains a valuable adjunct in modern regenerative oral surgery, with the potential to reduce reliance on extensive grafting procedures and enhance both functional and esthetic outcomes.

## 5. Conclusions

The results of this clinical study confirm that platelet-rich fibrin (PRF), when applied in combination with bone grafting materials and a resorbable membrane, significantly improves the effectiveness of guided bone regeneration in both horizontal and vertical defects of the alveolar ridge. Through its biological properties, PRF contributes to enhanced vascularization, supports tissue remodeling, and facilitates the long-term integration of dental implants. Clinically, the use of PRF is associated with reduced postoperative discomfort, improved healing of soft tissues, and increased implant stability. However, the findings also indicate that PRF used in isolation does not provide the mechanical structure required for successful regeneration in complex or large defects. Although it offers substantial biological benefits, PRF does not eliminate the need for particulate grafts or barrier membranes. The most favorable clinical outcomes are achieved when PRF is integrated into a broader regenerative protocol that includes both biologically active and structurally supportive materials. In this context, PRF serves as a valuable adjunct that enhances the healing environment without replacing conventional grafting techniques.

## Figures and Tables

**Figure 1 diagnostics-15-02078-f001:**
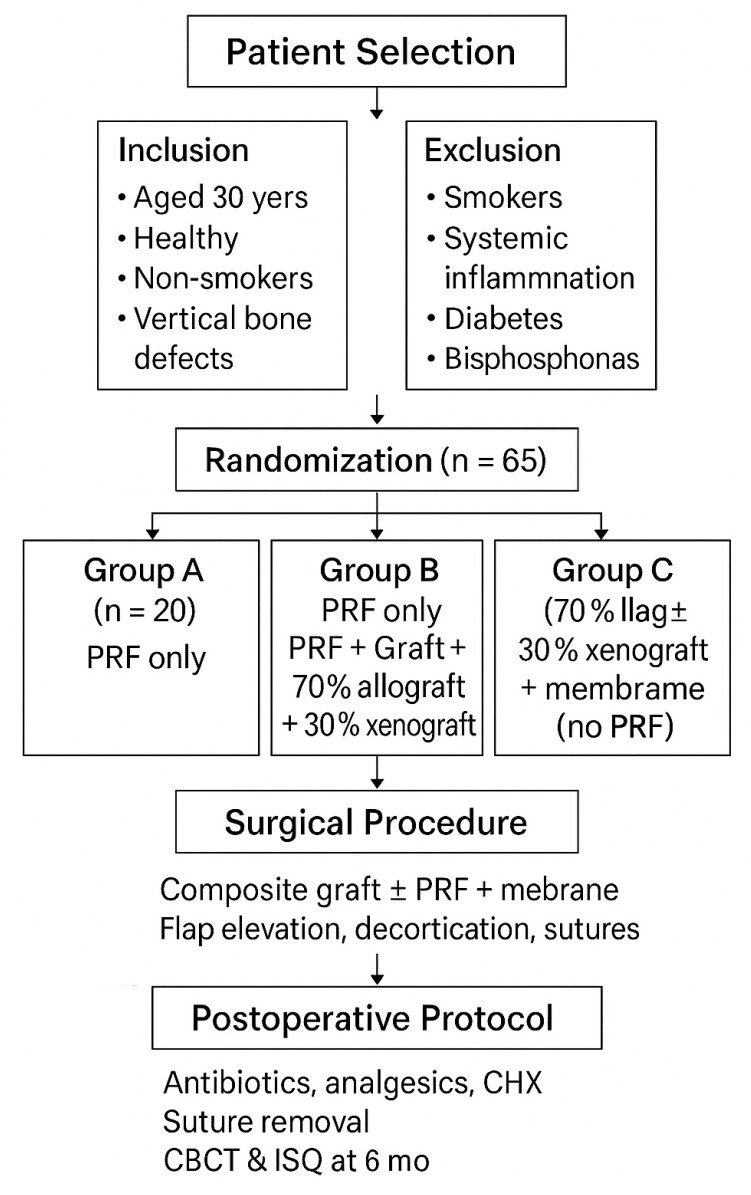
Study design and treatment workflow.

**Figure 2 diagnostics-15-02078-f002:**
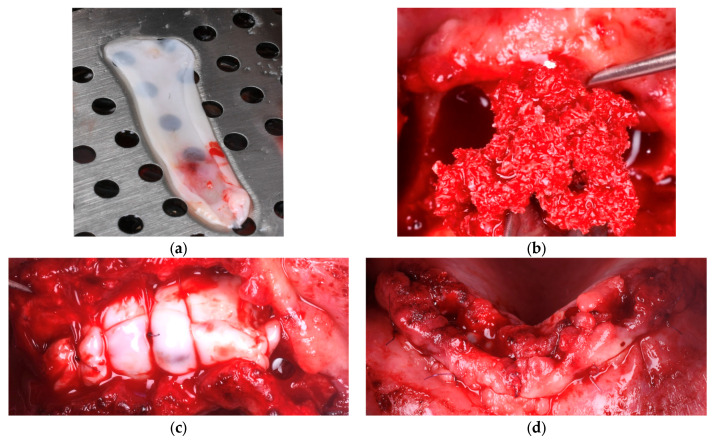
(**a**) Autologous PRF membrane prepared on a metal tray before placement. (**b**) Blend of 30% xenograft (Bio-Oss®, Geistlich Pharma AG, Switzerland) and 70% allograft (Puros Cancellous Particulate Allograft, ZimVie, Germany). (**c**) Multiple layers of PRF sutured over the defect and stabilized to the periosteum. (**d**) Final flap closure in Group A using horizontal mattress and single interrupted sutures to ensure stable healing.

**Figure 3 diagnostics-15-02078-f003:**
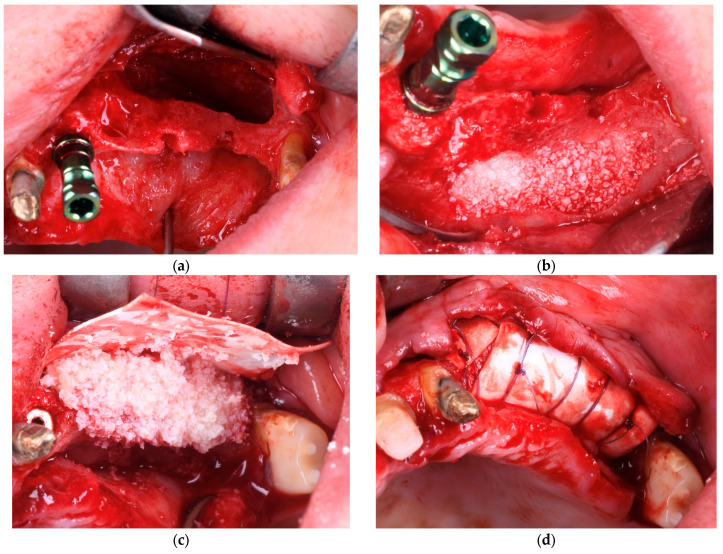
(**a**) Initial clinical view of the bone defect. (**b**) Placement of the bone graft composed of 30% xenograft (Bio-Oss®, Geistlich Pharma AG, Switzerland) and 70% allograft (Puros Cancellous Particulate Allograft, ZimVie, Germany). (**c**) Application of a resorbable collagen membrane (Biomed Extend, Zimmer Biomet, Warsaw, IN, USA) over the grafted area. (**d**) Stabilization of the grafted site using vertical mattress sutures with resorbable material (Chirmax 5-0 DS15, Chirmax, Prague, Czech Republic), prior to final positioning of the PRF membranes.

**Figure 4 diagnostics-15-02078-f004:**
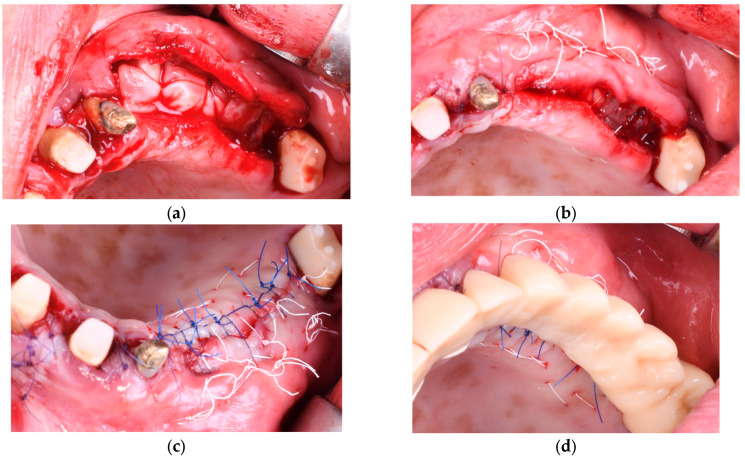
(**a**) Final positioning of PRF membranes before flap advancement with resorbable sutures (Chirmax 5-0 DS15, Chirmax, Prague, Czech Republic). (**b**) Tension-free flap adaptation over the grafted and membrane-covered area. (**c**) Suturing with a combination of horizontal mattress and single interrupted sutures. (**d**) Provisional restoration placed to maintain the space and protect the surgical site postoperatively.

**Figure 5 diagnostics-15-02078-f005:**
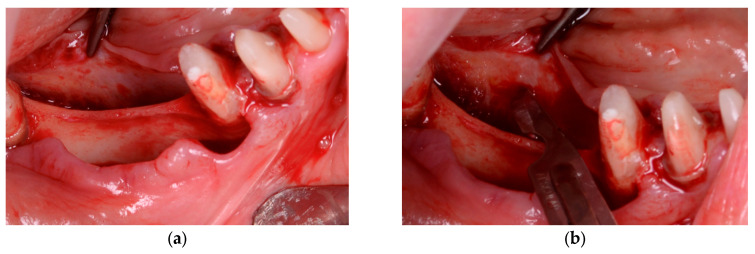
(**a**) A split-thickness flap was elevated by separating the mucosa from the periosteum to allow tension-free repositioning of the soft tissues. (**b**) The periosteum was incised to release the flap and increase its elasticity, facilitating passive adaptation over the grafted area after augmentation. (**c**) In this case with dense cortical bone, decortication was performed by creating multiple perforations with a 0.9 mm NTI round bur (Globus, Germany), facilitating medullary bleeding and improving cell recruitment and vascularization. (**d**) Positioning of the resorbable collagen membrane (Biomed Extend, Zimmer Biomet, Warsaw, IN, USA).

**Figure 6 diagnostics-15-02078-f006:**
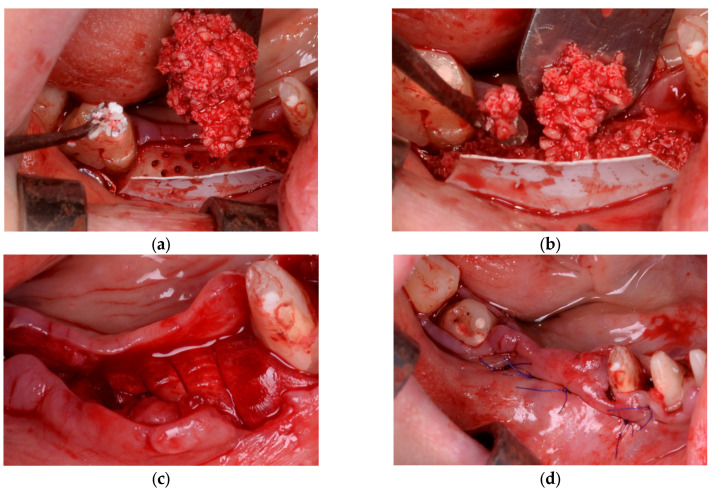
(**a**) Application of the bone graft composed of 30% xenograft (Bio-Oss®, Geistlich Pharma AG, Switzerland) and 70% allograft (Puros Cancellous Particulate Allograft, ZimVie, Germany). (**b**) Compaction of the grafting material into the defect site. (**c**) Placement of vertical mattress resorbable sutures (Chirmax 5-0 DS15, Chirmax, Prague, Czech Republic) prior to the final positioning of the resorbable collagen membrane (Biomed Extend, Zimmer Biomet, Warsaw, IN, USA). (**d**) Flap closure was achieved using horizontal mattress and single interrupted sutures with the same resorbable material to ensure tension-free adaptation and stable wound closure.

**Figure 7 diagnostics-15-02078-f007:**
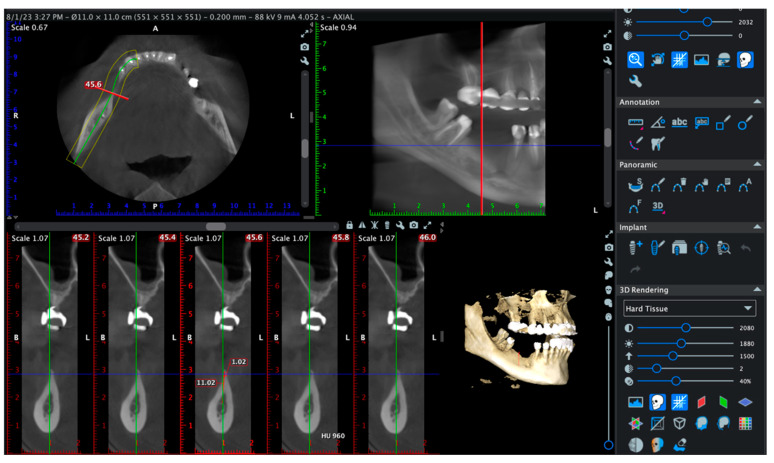
CBCT-based evaluation at 6 months postoperatively in a representative case. Cross-sectional analysis at 45.6 mm revealed a horizontal ridge width of 11.02 mm and a trabecular bone density estimated at 960 Hounsfield Units (HU). The buccal and lingual cortices were intact, with no evidence of dehiscence or fenestration. The 3D volumetric reconstruction confirmed complete bone coverage and spatial integrity around the radiopaque implant structure.

**Figure 8 diagnostics-15-02078-f008:**
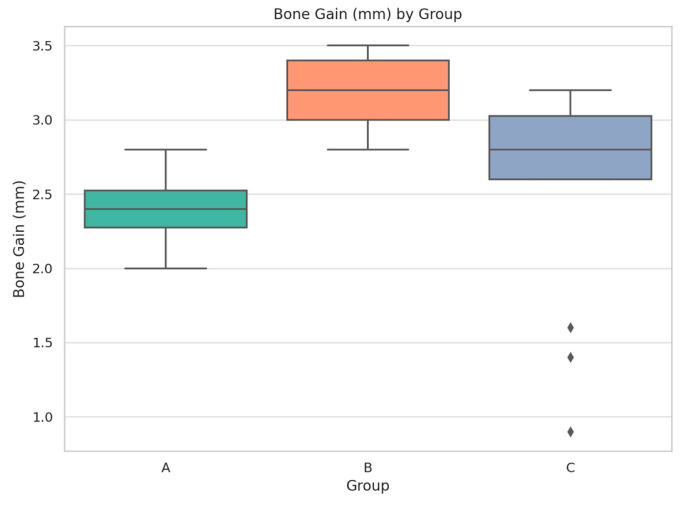
Comparative analysis of vertical bone gain following regenerative procedures in the three study groups.

**Figure 9 diagnostics-15-02078-f009:**
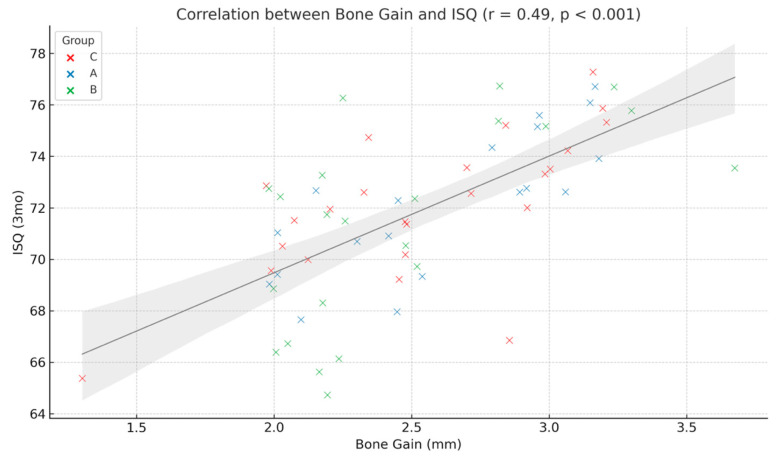
Correlation between vertical bone gain and implant stability (ISQ).

**Figure 10 diagnostics-15-02078-f010:**
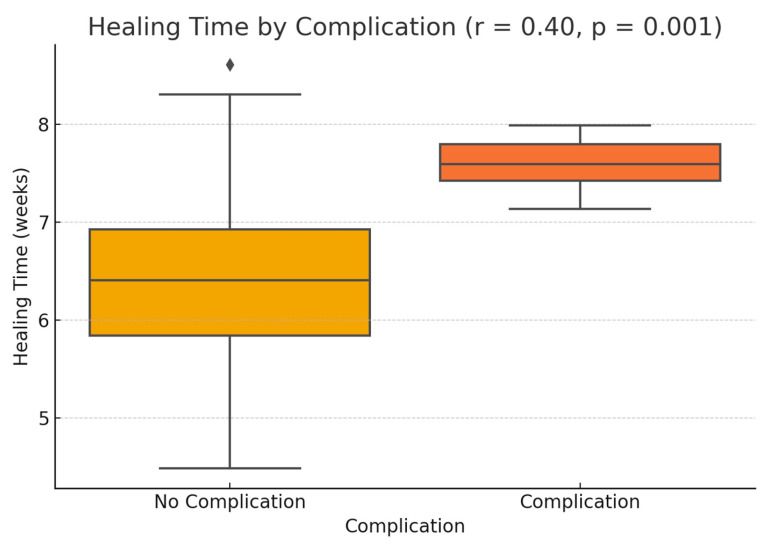
Distribution of healing time in patients with and without postoperative complications.

**Figure 11 diagnostics-15-02078-f011:**
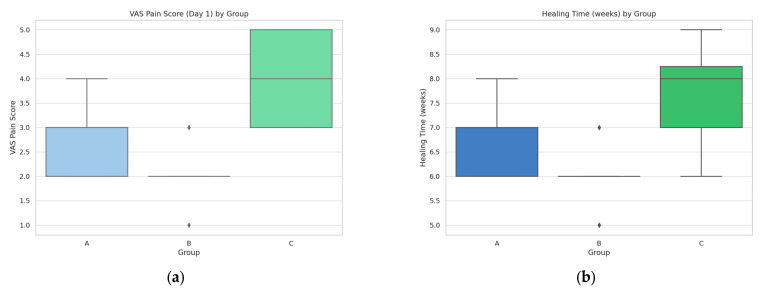
(**a**) VAS pain scores on postoperative day 1. (**b**) Healing time in patients with and without complications.

**Figure 12 diagnostics-15-02078-f012:**
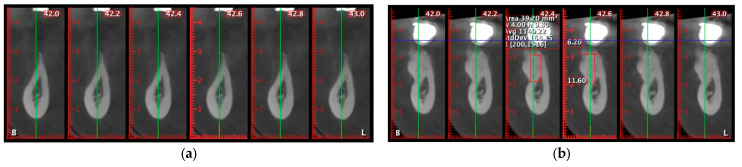
(**a**) Preoperative slice showing severe horizontal ridge deficiency (width: 1.20 mm). (**b**) Six-month postoperative slice showing horizontal ridge width increased to 11.63 mm and continuous corticalization, confirming successful bone regeneration and graft mineralization.

**Figure 13 diagnostics-15-02078-f013:**
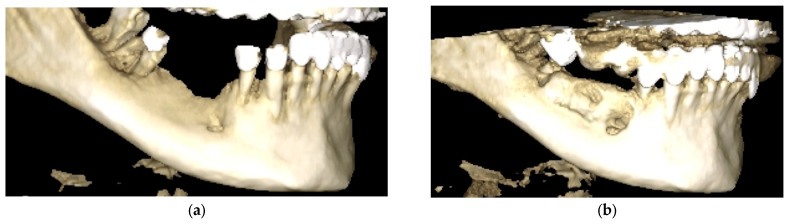
Three-dimensional CBCT reconstructions of the posterior mandible. (**a**) Preoperative rendering showing extensive alveolar resorption and cortical discontinuities. (**b**) Six-month postoperative reconstruction illustrating volumetric restoration and cortical continuity.

**Figure 14 diagnostics-15-02078-f014:**
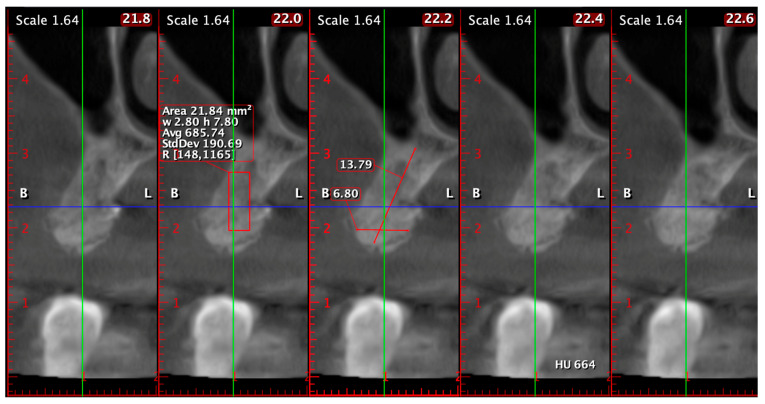
CBCT evaluation of the posterior maxilla at 6 months postoperatively: vertical ridge height (13.79 mm), bucco-palatal width (6.80 mm), and average trabecular density of 685.74 HU, indicative of mature regenerated bone suitable for implant insertion.

**Table 2 diagnostics-15-02078-t002:** PRF protocol used in the study and additional standardized variants.

Type	RPM/g-Force	Time (min)	Device/Notes
PRF (used in this study)	1500 RPM	14	Plasmalifting XC-2415; protocol used for clinical application
L-PRF	2700–3000 RPM	10–12	Intraspin; common standard for clot formation
A-PRF	1300 RPM	14	Advanced fibrin clot with high leukocyte content
A-PRF+	1300–1500 RPM	8	Equal platelet distribution; shorter protocol
I-PRF	700–800 RPM	3	White tubes; liquid PRF used before coagulation
C-PRF	2000 g	8	White tubes; high mononuclear cell concentration
PT Block	2700 RPM + 9 min	3 + 9	Intraspin; initial i-PRF, then L-PRF membrane from white tubes
PRP (Female)	3000 RPM	3	Sex-specific protocol (female); anticoagulated
PRP (Male)	3000 RPM	4	Sex-specific protocol (male); anticoagulated

**Table 3 diagnostics-15-02078-t003:** Overall patient demographics.

Variable	Value
Total patients	65
Male patients	34 (52.3%)
Female patients	31 (47.7%)
Mean age (years)	46.3 ± 7.8
Median age (years)	47
Age range (years)	30–64

**Table 4 diagnostics-15-02078-t004:** Patient demographics and group assignment.

Group	Description	Number of Patients (n)
Group A	PRF + Composite Graft (70% Allograft + 30% Xenograft)	20
Group B	PRF + Composite Graft + Collagen membrane	25
Group C	Composite Graft + Collagen Membrane (Control group)	20

**Table 5 diagnostics-15-02078-t005:** Evolution of implant stability (ISQ values) at placement and after 3 months across the three study groups.

Group	ISQ at Placement (Mean ± SD)	ISQ at 3 Months (Mean ± SD)
A	65.4 ± 3.8	71.5 ± 2.3
B	70.1 ± 3.1	74.2 ± 1.8
C	64.2 ± 4.0	68.9 ± 2.9

**Table 6 diagnostics-15-02078-t006:** Distribution of soft tissue complications and infection rates by study group. Statistically significant differences were observed (Chi-square test, *p* = 0.029).

Group	Total Patients (n)	Infection Cases (n)	Dehiscence Cases (n)	Graft Loss (n)	Infection Rate (%)	Dehiscence Rate (%)	Graft Loss Rate (%)
Group A (PRF only)	20	0	0	0	0	0	0
Group B (PRF + Graft + Membrane)	25	0	0	0	0	0	0
Group C (Control)	20	3	5	2	15	25	10

## Data Availability

Data are contained within the article.

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
