# Peer review of "Diagnostic and Clinical Outcomes of Three Regenerative Strategies for Alveolar Bone Defects: A Comparative Study Using CBCT and ISQ"

_diagnostics, 2025, doi:10.3390/diagnostics15162078_

Round 1
Reviewer 1 Report
Comments and Suggestions for Authors
I am not familiar with the regulations in Romania. Is it common for Ethics Committee of the University of Medicine and Pharmacy “Victor Babeș” of Timișoara, Romania to approve and oversee clinical treatments at a private dental clinic?
Subject randomization: Please describe details on how the patients were randomized.
"All radiographic measurements were performed by a calibrated examiner." Was the examiner blinded in terms of patient grouping? Did the examiner measure twice and what was the intra-examiner agreement?
Please report the scanning parameters (e.g. FOV, voxel size, ULD on/off) of the CBCT scans.
Figure 7: Please explain what the diamonds from Group C mean.
Figure 8: The SD bars are uninformative as they overlap with each other (baseline + at 3 months). The color key does not match with the actual figure.
Author Response
Dear Reviewer,
Thank you for your valuable comments, which help improve the clarity and methodological rigor of our manuscript. Please find below our point-by-point responses:
- “I am not familiar with the regulations in Romania. Is it common for Ethics Committee of the University of Medicine and Pharmacy ‘Victor Babeș’ of Timișoara, Romania to approve and oversee clinical treatments at a private dental clinic?”
Thank you for raising this point. In Romania, all clinical studies conducted by academic staff affiliated with a medical university must be reviewed and approved by the Ethics Committee of their respective institution, regardless of whether the clinical procedures are carried out in a university hospital or in a private medical or dental clinic.
In our case, four of the authors are academic staff members of the University of Medicine and Pharmacy “Victor Babeș” Timișoara. According to national regulations and internal university policy, any research initiated or coordinated by university faculty must be submitted for ethical approval to the university’s Ethics Committee. This ensures that the study adheres to the same ethical and scientific standards required for academic research, independent of the clinical setting.
It is, therefore, standard procedure in Romania for the Ethics Committee of the University of Medicine and Pharmacy “Victor Babeș” to approve and oversee research activities even if they are performed in private clinics, provided that the investigators are affiliated with the university.
We hope this clarifies the situation.
- “Subject randomization: Please describe details on how the patients were randomized.”
We have added the following text to theMaterials and Methods section to clarify both the randomization process and the blinding of the examiner:
“Patients were randomly allocated to the three study groups using a computer-generated randomization list prepared by an independent investigator who was not involved in the surgical procedures or outcome assessment. Group assignments were concealed in sequentially numbered opaque envelopes that were opened only at the time of surgery to ensure allocation concealment. All CBCT-based radiographic measurements and ISQ recordings were performed by a single calibrated examiner who was blinded to the treatment group allocation. To ensure reproducibility and minimize intra-observer variability, each radiographic measurement was repeated twice at a two-week interval. Intra-examiner agreement was calculated using the intraclass correlation coefficient (ICC), which demonstrated excellent reliability (ICC = 0.94).”
- “‘All radiographic measurements were performed by a calibrated examiner.’ Was the examiner blinded in terms of patient grouping? Did the examiner measure twice and what was the intra-examiner agreement?”
This request is addressed in the same newly added text quoted above, clarifying examiner blinding, repeated measurements, and ICC reliability. - “Please report the scanning parameters (e.g. FOV, voxel size, ULD on/off) of the CBCT scans.”
We have now included the following in theMaterials and Methods section:
“All CBCT scans were acquired using a standardized protocol with a field of view (FOV) of 5×5 cm for localized defects and 11×5 cm for extended ridge defects. The voxel size ranged between 150 μm and 200 μm depending on the selected FOV. Ultra-low dose (ULD) mode was activated for all scans to minimize radiation exposure. The acquisition time was approximately 4–5 seconds, with a tube voltage of 86–89 kV and a tube current of 6.3–8.0 mA.”
- “Figure 7: Please explain what the diamonds from Group C mean.”
In Figure 7, the diamonds represent outlier values within Group C. They are automatically generated by the statistical software used to create the boxplot to indicate individual measurements that fall outside the typical interquartile range. We have now clarified this in the revised figure legend. - “Figure 8: The SD bars are uninformative as they overlap with each other (baseline + at 3 months). The color key does not match with the actual figure.”
We agree that Figure 8 does not add substantial value beyond the numerical data already provided in Table 3 and the text. Moreover, the SD bars are uninformative due to their overlap, and the color key inconsistency may cause confusion. Therefore, we will remove Figure 8 from the manuscript to improve clarity and conciseness.
We trust that these clarifications address all your concerns.
Kind regards,
Reviewer 2 Report
Comments and Suggestions for Authors
Insufficient description of the mechanism of action of PRF.
Repetitive information.
There is no specific area where the procedure was performed.
No statistics on gender and age.
Insufficient literature.
In the introduction, the authors should describe the mechanism of action of the materials used in the study and cite more articles. The authors are repeating the same information. This is unnecessary. The topic of the paper is not groundbreaking, but it is interesting because of the comparison of different methods used in practice. The authors should describe in the paper the areas of the maxilla or mandible they performed the procedures. They should include a table describing patient data (age, gender, etc.), not just the number of patients participating in the experiment. In my opinion, the photos taken during the procedures are good, but a flow chart of the procedures would be useful. The results should include gender and the region in which the procedures were performed.
Author Response
Dear Reviewer,
Thank you for your detailed feedback, which helps refine and clarify our manuscript. Please find below our responses to each of your observations:
- “Insufficient description of the mechanism of action of PRF. In the introduction, the authors should describe the mechanism of action of the materials used in the study and cite more articles.”
We respectfully note that theIntroduction section already provides a detailed description of the biological and structural mechanisms of PRF. It explains the gradual release of growth factors (PDGF, TGF-β1, VEGF, IGF, FGF-2), their role in angiogenesis, cell recruitment, osteoblast proliferation, and differentiation, as well as the fibrin scaffold acting as a support for cell adhesion and migration. These mechanisms are supported by an extensive number of references (14–38).
To further improve the section, we removed redundant citations and added several new, more relevant references to strengthen the background on PRF’s mechanism of action and its regenerative potential. - “Repetitive information.”
We have carefully revised theIntroduction to remove repeated statements related to PRF’s biological effects and background. The text is now more concise and focused, without losing essential mechanistic details. - “There is no specific area where the procedure was performed. The authors should describe the areas of the maxilla or mandible they performed the procedures.”
We have clarified in theResults section that the procedures were performed in both the maxilla and mandible. Specifically, in the maxilla, 20 patients (30.8%) required interventions in the anterior region (incisors and canines), 10 patients (15.4%) in the premolar area, and 5 patients (7.7%) in the molar region. In the mandible, 8 patients (12.3%) presented indications in the anterior region, 12 patients (18.4%) in the premolar area, and 10 patients (15.4%) in the molar region. This anatomical distribution is now clearly stated. - “No statistics on gender and age. They should include a table describing patient data (age, gender, etc.), not just the number of patients participating in the experiment.”
We have already includedTable 3, which summarizes overall patient demographics (total number of patients, gender distribution, mean and median age, and age range). Additionally, Table 4 presents the demographic distribution across the three study groups. These tables provide a clear statistical overview of age and gender for all participants. - “A flow chart of the procedures would be useful.”
A flowchart illustrating patient selection, randomization, group allocation, surgical protocol, and follow-up is already provided asFigure 1, which covers this request. - “The results should include gender and the region in which the procedures were performed.”
This information is already included in theResults As mentioned above, gender distribution (34 males, 31 females) is provided, and the specific regions of the maxilla and mandible where the interventions were performed are described.
We trust that these clarifications demonstrate that the requested information has been addressed in the revised manuscript.
Kind regards,